# Effectiveness of Microcurrent Therapy for Treating Pressure Ulcers in Older People: A Double-Blind, Controlled, Randomized Clinical Trial

**DOI:** 10.3390/ijerph191610045

**Published:** 2022-08-15

**Authors:** Juan Avendaño-Coy, Noelia M. Martín-Espinosa, Arturo Ladriñán-Maestro, Julio Gómez-Soriano, María Isabel Suárez-Miranda, Purificación López-Muñoz

**Affiliations:** 1Faculty of Physiotherapy and Nursing of Toledo, University of Castilla-La Mancha, 45071 Toledo, Spain; 2Rest Home Montes de Toledo, 45460 Manzaneque, Spain; 3Nursing Home San Diego, 45600 Talavera de la Reina, Spain

**Keywords:** aged 80 and over, electric stimulation, electric stimulation therapy, nursing care, pressure ulcer wound healing

## Abstract

The aim of this study was to assess the effectiveness of microcurrent therapy for healing pressure ulcers in aged people. A multicentric, randomized clinical trial was designed with a sham stimulation control. The experimental group received an intervention following a standardized protocol for curing ulcers combined with 10 h of microcurrent therapy daily for 25 days. The sham group received the same curing protocol plus a sham microcurrent stimulation. The studied healing-related variables were the Pressure Ulcer Scale for Healing (PUSH) and the surface, depth, grade, and number of ulcers that healed completely. Three evaluations were conducted: pre-intervention (T1), 14 days following the start of the intervention (T2), and 1 day after the intervention was completed (T3). In total, 30 participants met the inclusion criteria (*n* = 15 in each group). The improvement in the PUSH at T2 and T3 was 16.8% (CI95% 0.5–33.1) and 25.3% (CI95% 7.6–43.0) greater in the experimental group versus the sham control, respectively. The reduction in the wound area at T2 and T3 was 20.1% (CI95% 5.2–35.0) and 28.6% (CI95% 11.9–45.3) greater in the experimental group versus the control, respectively. Microcurrent therapy improves the healing of pressure ulcers in older adults, both quantitatively and qualitatively.

## 1. Introduction

The prevalence of pressure ulcers is high in the older population [1], resulting in an important public health issue and negatively impacting individuals’ quality of life, health deterioration, pain, and risks of impairment and death [2,3,4]. Additionally, it entails elevated expenditure, with an estimated cost of USD 15,400 per person per year in public health systems [5]. Electric stimulation for treating pressure ulcers has proven to lower medical care expenditure, with a 51% reduction in clinician visits and an estimated annual saving of USD 519.6 per healed grade III/IV pressure ulcer [6].

The clinical practice guideline from the American College of Physicians recommends electric stimulation to accelerate the healing of pressure ulcers as adjuvant therapy [7]. High-voltage currents and direct currents are standard therapies for pressure ulcers delivered above the sensitivity threshold [6]. Microcurrents present a key differentiating factor: they deliver current intensities of microamps (µA) compared to milliamps (mA) in traditional electrical stimulation methods. Microcurrents are better tolerated by the patient since they remain below the sensitivity threshold without reaching the depolarization of nerve fibers [8]. Of note, the endogenous electric fields generated during natural wound healing, which play an essential role in cell migration and the healing of epithelial wounds, are in the order of microcurrents (≈5 µA/cm^2^) [9]. An in vitro study in human skin cells showed that microcurrents boosted the activity of fibroblasts and U937-monocytic cell lines, increasing the secretion of transforming growth factor beta-1 (TGF-ß1), an important mediator in inflammatory and regenerative responses, by 30% [10].

Clinical trials have evidenced a positive effect of microcurrents on the healing of chronic wounds [11] and diabetic ulcers [12,13,14]. A controlled randomized clinical trial (RCT) by Ullah showed that microcurrent therapy improved the healing of pressure ulcers in hospitalized patients, but the effect decreased with age [15]. However, this study did not specify the current parameters and dosage. A trial by Lessiani et al. delivering three microcurrent sessions for 30–40 min daily found the treatment to be effective, but they also did not report the current parameters [16].

The main aim of this trial was to assess the effectiveness of a standardized nursing-care protocol combined with microcurrent patches 10 h/day on the healing process of pressure ulcers versus the nursing-care protocol plus a sham stimulation in older adults residing in nursing homes. The secondary objectives were to analyze the effect of microcurrent therapy on the periulcer blood flow, arterial blood pressure, analgesics intake, capillary glycemia, and ulcer infection.

## 2. Materials and Methods

### 2.1. Study Design

A multicentric, parallel, double-blind, randomized clinical trial was designed with a placebo control. The Ethics Committee for Clinical Research of the Talavera de la Reina (Toledo, Spain) health area approved this study (Reference CEIC13/2018), which was registered before the recruitment of participants at Clinicaltrials.gov (Reference NCT0375358).

Participants were randomly allocated into two intervention groups (active microcurrents and sham) based on a randomized sequence created with Epidat 4.1. software (Sergas, Junta de Galicia, Santiago de Compostela, Spain). The therapists, participants, and evaluators measuring the outcome variables were blinded to the allocation. The devices used to deliver the active and sham stimulations were in the manufacturer’s packaging and looked identical. An independent researcher assigned participants to each group, giving a code for each patient and stimulator. Nurses from the included centers delivered the interventions according to the codes for the devices and electrodes.

### 2.2. Participants and Setting

Subjects were residents in nursing homes in the Toledo province who volunteered to participate, >65 years of age, with pressure ulcers. Ten nursing homes for older people agreed to participate, and care-providing nurses from the included centers recruited the participants from October 2019 to April 2021. Eligible volunteers were screened using the following inclusion criteria: pressure ulcer grades II, III, and IV according to the North American National Pressure Ulcer Advisory Panel System and the European Pressure Ulcer Advisory Panel System (NPUAP/EPUAP) [17]; evolution time 1–24 months; ulcer area >1 cm^2^; and <14 points on the Braden scale for pressure ulcer risk [18]. The criteria for exclusion were: a cardiac pacemaker or another implanted electric device, osteosynthesis implants near the ulcer, pressure ulcers in the occipital region, cancer, osteomyelitis, ≥3 abnormal blood markers at baseline indicating limited healing potential (anemia, iron deficit, protein deficit, dehydration, non-controlled diabetes, or hypothyroidism), allergy to the usual treatment for ulcer healing, systemic infection, and ulcer treatment with growth factor or vacuum-assisted closure in the 30 days before the trial. All participants and relatives were informed about the trial, verbally and in written form, and provided consent before their inclusion. A relative or legal tutor provided consent when participants were incapable of making decisions. The trial complied with the ethical principles of the Helsinki Declaration for medical research in humans [19].

### 2.3. Intervention

The experimental group received a standardized nursing-care protocol combined with active microcurrent patches. The standardized protocol for healing ulcers followed a dynamic strategy named TIME that summarizes the four key points for stimulating the natural repair process [20]: T (Tissue)—the control of non-viable tissue and the debridement of necrotic tissue; I (Infection)—the control of local inflammation and infection; M (Moisture)—the control of exudate and cures in a humid environment; and E (Edge)—the stimulation of epithelial edges. This protocol followed the “Guide to recommendations based on evidence in prevention and treatment of pressure ulcers in adults” by the Osakidetza health service (Spain) [21].

In the microcurrent intervention, two 10 cm^2^ electrodes (Mc Patch, Newmark Inc., Manhattan, NY, USA) were applied around the ulcer at the edge of the dressing, 10 h/day for 25 consecutive days or until the complete healing of the wound. The characteristics of the delivered microcurrents were: a monophasic, pulsed, square-form wave pulse of 1.5 s with a 300 ms pause, a voltage of 21 mV, the intensity of 42 µA, and the current density of 4.2 µA/cm^2^.

The sham group received the same nursing-care protocol, but the device was manipulated and tested with an oscilloscope to ensure that no electric current would be emitted.

The success of the allocation blinding of therapists and evaluators was assessed at the end of the intervention.

### 2.4. Variables

Demographic and clinical variables were collected at baseline. The recorded demographic characteristics were: age (years), gender, and weight (kg). The clinical variables were: time since the ulcer onset, location, ulcer grade according to the NPUAP/EPUAP (grades II–IV) [17], infection (exudate culture: positive), pressure ulcer risk according to the Braden scale [18], diabetes, the use of an anti-bedsore mattress, protein supplements, cognitive condition assessed with the Spanish version of Mini-Mental Status Examination [22], comorbidity (number of diseases), systolic and diastolic blood pressure (mmHg), radial pulse, and capillary glycemia (mg/dL).

Outcome variables were measured at three time points: before the intervention (T1), 14 days following the start of the intervention (T2), and one day after the completion of the intervention (T3). The main outcome variable was ulcer healing measured with the Pressure Ulcer Scale for Healing (PUSH) [23] that categorizes ulcers by assessing three parameters both quantitatively and qualitatively: the length x width of the ulcer (measuring the greatest length—head to toe—and the greatest width—side to side—using a centimeter ruler. It scores from 0—surface area of 0 cm^2^—to 10—surface area >24 cm^2^); exudate amount (classified as none: score 0; light: score 1; moderate: score 2; or heavy: score 3); and tissue type (score 1: superficial wound with epithelial tissue; score 2: the wound is clean and contains granulation tissue; score 3: there is any amount of slough present; score 4: necrotic tissue is present. If the wound is closed, the score is 0). Each of the parameters described has a sub-score, and by adding the three sub-scores, the total score is obtained. The total score ranges from 0 (completely healed) to 17 (the worst state of the wound). A comparison of total scores measured in each time point provides an indication of the improvement or deterioration in pressure ulcer healing. Other healing-related variables were: wound area (cm^2^), measured with the photograph analysis app Mobile Wound Analyzer (MOWA) (Healthpath, London, UK); ulcer depth (mm), measured with a sterile swab at the deepest point; ulcer grade; and the number of completely healed ulcers.

Other evaluated secondary variables were: blood flow in the area surrounding the ulcer, measured via laser Doppler flowmetry on a 0–1000 unit scale (model DRT4, Moor Instruments, Devon, UK) [24]; analgesics intake and changes versus baseline; wound infection, for which an exudate culture was performed when infection was suspected; capillary glycemia (mg/dL) calculated as the average of two measurements 24 h apart; and systolic and diastolic pressure (mm/Hg) and radial pulse at rest, estimating the average of two measurements taken 1–2 min apart.

For the blinding assessment, therapists and evaluators were questioned separately about the treatment assignment after the last session via a close-ended questionnaire, following the recommendations by Bang et al. and Kolahi et al. [25,26].

### 2.5. Statistical Analysis

Sample size was estimated using the Epidat 4.1 software based on the results of a former study that delivered microcurrent therapy to patients with pressure ulcers and where the PUSH score was the main outcome variable [16]. A minimum of 9 participants per group was obtained (*n* = 18) for a confidence level of 95% and a power of 85%. Considering losses to follow-up, the final sample size was 30 participants (*n* = 15 in each group).

An intention-to-treat analysis was conducted, and the last available measurement was employed when data were missing due to dropouts. For the intergroup comparison of basal characteristics, a descriptive analysis and inferential statistics for basal demographic and clinical variables were performed for independent groups (parametric or non-parametric depending on the variable). A two-factor (intervention-time) repeated-measures analysis of variance (ANOVA) with Bonferroni post hoc correction was conducted for the following outcome variables: PUSH score, area, depth, periulcer skin blood flow, glycemia, blood pressure, and pulse. The Greenhouse–Geisser correction was employed for variables violating sphericity. The Student’s *t*-test for independent samples was used to assess changes in intergroup comparisons. Changes over time in ulcer grade were analyzed with the Friedman test and Mann–Whitney *U*-test for intergroup comparisons. Fisher’s exact test was employed to evaluate the intervention effect on both analgesics intake and infection.

To analyze blinding success, James’ Blinding Index (BI) [27] and Bang’s Blinding Index [25] were obtained using Stata v15.0 software (StataCorp., College Station, TX, USA). James’ BI infers the overall blinding success in all arms. It ranges from 0 to 1 (0 = total lack of blinding, 1 = complete blinding, 0.5 = completely random blinding). Bang’s BI characterizes and evaluates the blinding in each arm independently. It ranges between −1 and 1, with 0 indicating the most desirable situation [26].

## 3. Results

A total of 34 participants were assessed for eligibility, 30 of which met the inclusion criteria and were randomly allocated into two intervention groups: active microcurrents (*n* = 15) and sham (*n* = 15). Three participants, one in the intervention and two in the placebo groups, passed away between T1 and T2 due to comorbidity and therefore were lost to follow-up. Finally, 30 subjects were included in the statistical analysis, because the three dropouts were analyzed using an intention-to-treat analysis in order to avoid the risk of bias, preserving the benefits of the initial randomization (Figure 1). No adverse effects related to the intervention were registered.

### 3.1. Demographic and Clinical Characteristics of Participants at Baseline

Subjects were recruited from seven of the ten included nursing homes, with no differences in the ratio of participants in each arm from each center (Fisher’s exact test = 3.2; *p* = 1.0). Table 1 shows the basal demographic and clinical characteristics of the sample. No intergroup differences were observed except for the Braden scale and the ratio of diabetic participants. The pressure ulcer grades were II, III, and IV in *n* = 6 (20.0%), *n* = 18 (60.0%), and *n* = 6 (20.0%) subjects, respectively, without intergroup differences (*p* = 0.26). A total of 23 (76.7%) participants (*n* = 11 in the experimental group and *n* = 12 in the sham group) showed moderate or severe cognitive deterioration (≤19 points in the Mini-Mental state examination test). Pressure ulcers were located in the sacral area (33.3%; *n* = 5 in the experimental group and *n* = 5 in the control group), calcaneal area (36.7%; *n* = 7 in the active group and *n* = 4 in the sham group), trochanter (16.7%; *n* = 3 in the active group and *n* = 2 in the sham group), lateral malleolus (10%, *n* = 3 in the sham group), and ischium (3.3%, *n* = 1 in the sham group), without intergroup differences (Fisher’s exact test, *p* = 0.33).

### 3.2. Effect on Pressure Ulcer Healing

Table 2 shows the outcomes of the PUSH scale, ulcer area, and ulcer depth. Changes were observed in the main outcome variable (PUSH score), in the time factor (ANOVA *F* = 17.5; *p* < 0.001), and the intersection time-intervention group (ANOVA *F* = 6.0; *p* = 0.004). The improvement in the PUSH at T2 and T3 was 16.8% (CI95% 0.5–33.1) and 25.3% (CI95% 7.6–43.0) greater in the experimental group versus the sham (Table 2). Statistically significant changes in ulcer areas were also observed in the time factor (ANOVA *F* = 9.2, *p* < 0.001) and the intersection time-intervention group (ANOVA *F* = 7.3, *p* = 0.002). The wound area reduction at T2 and T3 was 20.1% (CI95% 5.2–35.0) and 28.6% (CI95% 11.9–45.3) greater in the experimental group versus the sham (Table 2). No differences in wound depth were observed in the time factor (ANOVA *F* = 3.5, *p* = 0.055) or the intersection time-intervention group (ANOVA *F* = 0.1, *p* = 0.81). The grade of pressure ulcers improved with time (Friedman test *p* < 0.001). Although no intergroup differences were observed at T1 (mean ranks: experimental = 13.9 vs. sham = 17.1; *p* = 0.26), a significantly greater improvement in ulcer grade was observed in the experimental group at T2 (mean ranks: experimental = 12.5 vs. sham = 18.5; *p* = 0.045) and T3 (mean ranks: active = 12.2 vs. sham = 18.8; *p* = 0.027). Complete healing occurred in *n* = 3 (20.0%) participants in the active group versus none in the sham group, although statistical significance was not reached (Fisher’s exact test *p* = 0.22).

### 3.3. Effect on Secondary Variables

Table 3 displays the outcomes of quantitative secondary variables. No intergroup or over-time differences were found in periulcer blood flow, blood pressure, or capillary glycemia. On the contrary, radial pulse showed a significant change in the time factor (ANOVA *F* = 7.4, *p =* 0.001). The post hoc analysis showed a decrease of 7.2 beats/min (CI95% 2.1–12.3) in the pulse rate at T3 in the sham group compared to baseline. In terms of prescribed painkillers, *n* = 18 participants (*n* = 8 (53.3%) in the experimental group and *n* = 10 (66.6%) in the sham group) were taking analgesics before the intervention, without significant intergroup differences (χ^2^ = 0.6, *p =* 0.46). None of the participants increased the prescribed dose, while 7 participants even lowered it (*n* = 4 (26.6%) in the experimental group and *n* = 3 (20.0%) in the sham group), with no intergroup differences (Fisher’s exact test *p* = 1.0). Infection was present in 3 participants at baseline (*n* = 2 and *n* = 1 in the sham and experimental groups, respectively), with no intergroup differences (Fisher’s exact test *p* = 1.0). One newly developed infection only appeared in the sham group at T2 and T3, without intergroup differences (Fisher’s exact test *p* = 0.60).

### 3.4. Blinding Assessment

The guesses of therapists and assessors about group allocations are shown in Table 4, as well as James’ and Bang’s BI in Table 5. The evaluation of participants’ blinding was not possible as we were not able to obtain responses about group allocation due to cognitive deterioration.

## 4. Discussion

The multidimensional PUSH scale scores and ulcer areas improved 25% and 29% more at T3 in the microcurrents group compared to the sham, respectively. These results are very similar to those reported by Lessiani et al., who observed greater improvements of 33% in the PUSH and 25% in the wound area in the experimental group versus the sham [16]. That study delivered the interventions for 90–120 min/day but did not include ulcers graded IV, and the average age (73 years) was lower. A multicentric RCT by Ullah comparing the effect of microcurrents versus a sham over 12 weeks also observed that the rate of change in the area of pressure ulcers was greater in the experimental group [15]. However, the age range (60–79 years) and average age (69.3 years, SD = 6.2) [15] were also inferior to the current trial (73–97 years; mean = 87.6, SD = 5.7). Additionally, aging appears to be a key factor in view of the results by Ullah, which showed its negative influence on the healing of pressure ulcers [15]. The two latter trials did not specify any additional therapy parameter [15,16]. The positive outcomes observed in the current trial, conducted in an older population, could be due to the longer application period (10 h/day). Since endogenous microcurrents continuously flow through the wound, a greater application time could be more effective. Our findings are in line with the results of a recent systematic review with metanalysis which determined that electrical microcurrent therapy may improve wound scarring by reducing the wound area, although this evidence is moderate [28]. However, there is no consensus on the parameters of electrical stimulation to be applied (current intensity, frequency, wave form, duration…), as there is heterogeneity in the studies and these parameters are often not specified. For this reason, further comparative trials with different application parameters are required to confirm this hypothesis.

This study did not find intergroup differences in the wound depth. Conversely, a significant improvement in ulcer grades was found in the experimental group compared to the sham. Additionally, 20% of ulcers attained complete healing in the microcurrents group versus none in the sham, although statistical significance was not reached (*p* = 0.22), possibly due to a short follow-up. A trial with a longer follow-up observed greater and more significant differences [29]. A case study reported the complete healing of a pressure ulcer on the sacral area 6 weeks after delivering microcurrents with the same device employed in the current trial [30]. Future clinical trials should establish a more prolonged follow-up period than ours, independently of the intervention duration.

Lee et al. conducted a cases series study in people with diabetic ulcers and reported that microcurrents improved glycemia and blood pressure levels, attributing this effect to the presumed antioxidant effect of microcurrents [13]; nevertheless, no intergroup differences were observed in these two outcome variables. Intergroup differences were also not found in wound infection, despite the potential bactericidal effect of microcurrents [31]. A trial in burn patients reported statistically significant differences in the bacterial count within wounds, with a 0.04% decrease in the microcurrents group versus an 86% increase in the control group [32]. The low rate of infected ulcers in the present trial did not allow conclusions to be drawn about the effect on this variable. Exudate culture was only performed in case of the clinical suspicion of infection and not systematically, so the number of infected ulcers could have been underestimated. This trial was designed to assess healing-related variables, so further research should evaluate the effect of microcurrents on them.

Periulcer blood flow did not show intergroup differences. Polak et al. also did not find differences at 4 weeks post-intervention. However, the latter applied high-voltage electric stimulation to pressure ulcers in people with neurological injuries [33]. Self-perceived pain could not be evaluated given the cognitive state of the participants, although the previously registered protocol included it. Analgesics intake was employed as an indirect measure, but no intergroup differences were found. Given the high cognitive deterioration experienced by participants, the blinding of participants can be considered to have been achieved. The blinding of evaluators and therapists was satisfactory overall, even if therapists believed that 40% of subjects in the sham group were allocated to the experimental group.

This study has limitations. Firstly, the high number of included nursing homes could have produced biases due to differences in the nursing care delivered, although the nursing protocol for wound care was standardized. Secondly, 53% of participants in the experimental group had diabetes at baseline versus 13% in the placebo group. Taking into account the negative impact of diabetes on the healing process [34], the effectiveness of microcurrents in healing pressure ulcers could have been underestimated.

## 5. Conclusions

In conclusion, microcurrent therapy in older people improved the healing of pressure ulcers, both quantitatively and qualitatively, without adverse effects. It is necessary to find additional treatments to improve patients’ quality of life and decrease morbidity and mortality associated with pressure ulcers in aged people. Comparative clinical trials should be designed to determine the optimal parameters of microcurrent electric stimulation for healing pressure ulcers.

## Figures and Tables

**Figure 1 ijerph-19-10045-f001:**
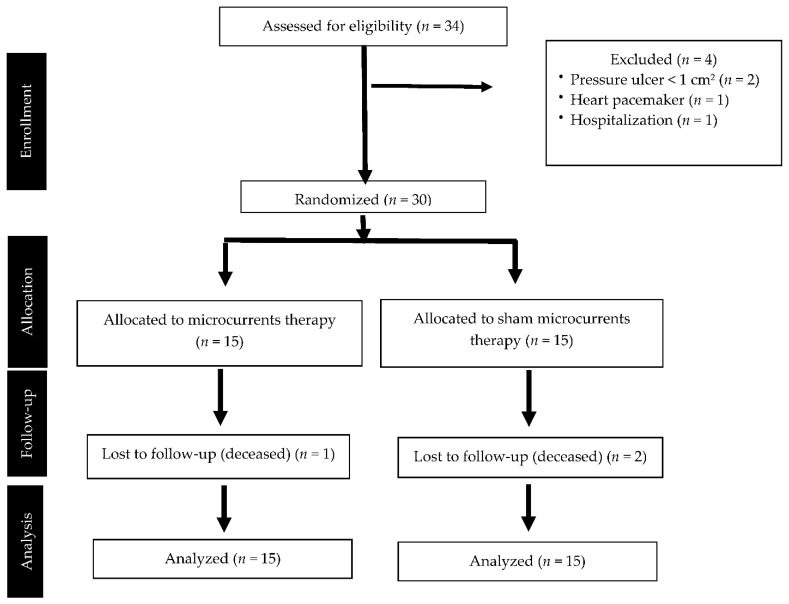
Flow chart of the clinical trial following the CONSORT guidelines.

**Table 1 ijerph-19-10045-t001:** Demographic and clinical characteristics of participants at baseline.

Outcomes	Participants (*n* = 30)	Microcurrents Group (*n* = 15)	Sham Group (*n* = 15)	Intergroup Differences (*p* Value)
Age (years) Mean (SD)	87.6 (5.7)	88.8 (5.0)	86.3 (6.2)	(*p* = 0.24) ^a^
Gender (Men/Women) *n* (%)	7 (23.3%)/23 (76.7%)	3 (20.0%)/12 (80.0%)	4 (26.7%)/11 (73.3%)	(*p* = 1.0) ^c^
Weight (kg) Mean (SD)	63.5 (14.1)	64.6 (15.3)	62.5 (13.2)	(*p* = 0.69) ^a^
PU duration (days) Mean (SD)	62.8 (63.7)	60.3 (38.6)	65.3 (83.2)	(*p* = 0.84) ^a^
PU grade (II–IV) Median/Mode	3/3	3/3	3/3	(*p* = 0.26) ^d^
PU area (cm^2^) Mean (SD)	7.4 (6.7)	5.2 (4.6)	9.5 (8.0)	(*p* = 0.09) ^a^
PUSH scale Mean (SD)	11.2 (2.5)	10.7 (2.6)	11.7 (2.4)	(*p* = 0.25) ^a^
PU infection Yes/No *n* (%)	3 (10.0%)/27 (90.0%)	1 (6.7%)/14 (93.3%)	2 (13.3%)/13 (86.7%)	(*p* = 1.0) ^c^
Braden scale Mean (SD)	11.6 (2.2)	12.5 (1.6)	10.8 (2.5)	**(*p* = 0.04) *^a^**
Diabetes Yes/No *n* (%)	10 (33.3%)/20 (66.7%)	8 (53.3%)/7 (46.7%)	2 (13.3%)/13 (86.7%)	**(*p* = 0.02) *^b^**
Anti-decubitus mattress Yes/No *n* (%)	15 (50.0%)/15 (50.0%)	7 (53.3%)/8 (46.7%)	8 (53.3%)/7 (46.7%)	(*p* = 0.72) ^b^
Protein supplements Yes/No *n* (%)	10 (33.3%)/20 (66.7%)	3 (10.0%)/12 (90.0%)	7 (46.7%)/8 (53.3%)	(*p* =0.12) ^b^
Comorbidity (number of diseases) Mean (SD)	3.4 (0.9)	3.5 (0.8)	3.3 (1.0)	(*p* = 0.55) ^a^
Mini-mental test Mean (SD)	12.5 (9.4)	12.5 (9.5)	12.4 (9.7)	(*p* = 0.97) ^a^
Systolic blood pressure (mmHg) Mean (SD)	124.8 (15.1)	124.0 (17.4)	125.6 (13.0)	(*p* = 0.78) ^a^
Diastolic blood pressure (mmHg) Mean (SD)	70.4 (11.0)	69.3 (9.9)	71.6 (12.1)	(*p* = 0.57) ^a^
Radial pulse (beats/min) Mean (SD)	79.2 (10.4)	76.1 (11.5)	82.2 (8.5)	(*p* = 0.11) ^a^
Blood glucose (mg/dL) Mean (SD)	99.8 (22.5)	103.8 (21.7)	95.8 (23.3)	(*p* = 0.34) ^a^
Periulcer flowmetry Mean (SD)	76.0 (50.5)	68.8 (36.3)	85.6 (67.7)	(*p* = 0.60) ^a^

Abbreviations: PU: pressure ulcer, PUSH: Pressure Ulcer Scale for Healing. Statistical tests: (^a^) Student’s *t*-test for independent samples, (^b^) Pearson’s chi-squared test, (^c^) Fisher’s exact test, (^d^) Mann–Whitney *U*-test. (*) Bold font indicates statistical significance (*p* < 0.05).

**Table 2 ijerph-19-10045-t002:** Results of variables related to the healing of pressure ulcers: intragroup and intergroup comparison of changes versus baseline.

Outcomes	Intragroup Comparison Versus Baseline				Intergroup Comparison of Changes versus Baseline	
	Active Group		Sham Group		Mean Change Active Minus Sham at T2	Mean Change Active Minus Sham at T3
	T1 Minus T2	T1 Minus T3	T1 Minus T2	T1 Minus T3		
PUSH scale % Mean (CI95%)	**22.5% **** **(8.5–36.5)**	**34.4% **** **(19.0–49.7)**	5.7% (−8.3–19.6)	9.1% (−6.3–24.5)	**16.8% *** **(0.5–33.1)**	**25.3% **** **(7.6–43.0)**
Pressure ulcer area % Mean (CI95%)	**22.0% **** **(8.9–35.1)**	**30.2% **** **(15.5–44.1)**	1.9% (−11.2–15.0)	1.6% (−13.1–16.2)	**20.1% *** **(5.2–35.0)**	**28.6% **** **(11.9–45.3)**
Pressure ulcer depth % Mean (CI95%)	12.7% (−1.2–26.6)	19.5% (−8.1–47.1)	6.4% (−7.5–20.3)	15.0% (−12.6–42.6)	6.3% (−9.5–22.2)	4.5% (−26.9–35.8)

Abbreviations: T1: pre-intervention, T2: during treatment at 14 days since beginning, T3: 1 day post-intervention at 26 days since beginning, PUSH: Pressure Ulcer Scale for Healing. Bold font indicates statistical significance: (******) *p* < 0.01; (*****) *p* < 0.05.

**Table 3 ijerph-19-10045-t003:** Results of quantitative secondary variables: intra-group and intergroup comparisons versus baseline.

Outcomes	Intragroup Comparison versus Baseline				Intergroup Comparison of Changes versus Baseline	
	Active Group		Sham Group		Mean Change Active Minus Sham at T2	Mean Change Active Minus Sham at T3
	T1 Minus T2	T1 Minus T3	T1 Minus T2	T1 Minus T3		
Periulcer flowmetry (%) Mean (CI95%)	−11.4% (−64.1–41.3)	24.8% (−20.2–69.8)	4.4% (−56.4–65.3)	29.3% (−22.7–81.2)	−15.8% (−78.9–47.3)	−4.5% (−58.3–49.4)
Systolic blood pressure (mmHg) Mean (CI95%)	1.2 (−6.9–9.3)	1.3 (−8.0–10.6)	1.7 (−6.4–9.8)	5.4 (−3.8–14.7)	−0.5 (−9.8–8.8)	−4.1 (−14.7–6.4)
Diastolic blood pressure (mmHg) Mean (CI95%)	−3.9 (−9.5–1.7)	−0.8 (−7.7–6.1)	−1.3 (−6.9–4.3)	2.8 (−4.1–9.8)	−2.6 (−9.0–3.8)	−3.6 (−11.5–4.3)
Radial pulse (beats/min) Mean (CI95%)	1.7 (−4.8–8.2)	5.0 (−0.1–10.1)	4.1 (−2.4–10.6)	**7.2 **** **(2.1–12.3)**	−2.4 (−9.8–5.0)	−2.2 (−8.0–3.6)
Blood glucose (mg/dL) Mean (CI95%)	−4.6 (−15.7–6.6)	−12.3 (−34.1–9.4)	0.7 (−10:4–11.9)	5.3 (−16.5–27.1)	−5.3 (−18.0–7.4)	−17.6 (−42.4–7.1)

Abbreviations: T1: pre-intervention, T2: during treatment at 14 days since beginning, T3: 1 day post-intervention at 26 days since beginning. Bold font indicates statistical significance: (******) *p* < 0.01.

**Table 4 ijerph-19-10045-t004:** Guesses of therapists (upper) and assessors (lower) about group allocations. MCT: microcurrent therapy.

Allocation	Therapist Guess, *n* (%)			
	Active MCT	Sham MCT	Do Not Know	Total
Active MCT	7 (23.3)	3 (10.0)	5 (16.7)	15 (50.0)
Sham MCT	8 (26.7)	2 (6.7)	5 (16.7)	15 (50.0)
Total	15 (50.0)	5 (16.7)	10 (33.3)	30 (100.0)
**Allocation**	**Assessor Guess, *n* (%)**			
	**Active MCT**	**Sham MCT**	**Do Not Know**	**Total**
Active MCT	5 (16.7)	2 (6.7)	8 (26.7)	15 (50.0)
Sham MCT	5 (16.7)	3 (10.0)	7 (23.3)	15 (50.0)
Total	10 (33.3)	5 (16.7)	15 (50.0)	30 (100.0)

**Table 5 ijerph-19-10045-t005:** James’ Blinding Index and Bang’s Blinding index of therapists (upper) and assessors (lower).

Methods	Index	*p*-Value	95% Confidence Interval	Conclusion
James’s BI	0.70	1.00	0.58 to 0.82	Blinded
Bang’s BI—Active/2 × 3	0.27	0.09	−0.06 to 0.59	Blinded
Bang’s BI—Sham/2 × 3	−0.40	0.99	−0.70 to −0.09	Opposite guess
James’s BI	0.73	1.00	0.60 to 0.85	Blinded
Bang’s BI—Active/2 × 3	0.20	0.12	−0.08 to 0.48	Blinded
Ban’s BI g—Sham/2 × 3	−0.13	0.76	−0.43 to 0.17	Blinded

## Data Availability

Original datasets are available in Zenodo repository at DOI:10.5281/zenodo.6990652.

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
