# Peer review of "Effectiveness of Microcurrent Therapy for Treating Pressure Ulcers in Older People: A Double-Blind, Controlled, Randomized Clinical Trial"

_ijerph, 2022, doi:10.3390/ijerph191610045_

Round 1

Reviewer 1 Report

This is a well developed clinical study on the effect of microcurrent therapy on pressure ulcers.  The methods and results are captured in remarkable detail providing a valuable resource for other studies in this area.  The introduction and discussion do a great job of putting the study in context with previous studies.  I would have liked more detail in the discussion comparing the stimulation parameters with previous studies, but it sounds like that information has not usually been reported in as much detail as provided in this publication.  

Author Response

Please, see the attachment file.

Reviewer 2 Report

Effectiveness of microcurrent therapy for treating pressure ulcers in older people: a double-blind, controlled, randomized clinical trial.

Thanks for the opportunity to review this paper.

I made some comments and tried to be as didactic as possible in my review.

I hope that my contribution represents a time of learning for all the people involved in this process (including myself).

GLOBAL OPINION:

The theme is quite interesting and current.

The manuscript is well written.

The main objective is clear.

The methodology is OK (please explain how you analyzed 15 participants in each group as there were 3 dropouts).

The results are clearly presented (Please review the figure 1, table 1 and table 2 formatting).

The conclusions are supported by the results/discussion.

The references are current and relevant.

SPECIFIC SUGGESTIONS:

Abstract/Keywords

I suggest using MeSH Terms

Available at https://meshb.nlm.nih.gov/

Introduction

Despite the PUSH being an internationally known scale, I suggest a brief description of it (and the different dimensions) in introduction and/or methodology (since it will be a "parameter" for comparison between the two groups).

Figure 1.

The authors state that there is 1 dropout in the test group and 2 dropouts in sham group (those 3 patients died during the study).

Although the authors analyzed 15 participants in each group. Please clarify/explain!

FORMATTING SUGGESTIONS:

To improve the analysis of figure 1, the flow chart should appear on the same page.

To improve the analysis of Table 1, the data should appear on the same page.

To improve the analysis of Table 2, the data should appear on the same page.
